



# PL1GD-T - gridded dataset of the mean, minimum and maximum daily air temperature at the level of 2 m for the area of Poland at a resolution of 1 km × 1 km

Adam Jaczewski[1], Michał Marosz[1], Mirosław Miętus[1,2]

[1]Institute of Meteorology and Water Management – National Research Institute, Podleśna 61, 01-673 Warsaw, Poland

[2]University of Gdańsk, Department of Physical Oceanography and Climate Research, Bażyńskiego 4, 80-309 Gdańsk, Poland

*Correspondence to*: Adam Jaczewski (Adam.Jaczewski@imgw.pl)

**Abstract.** This paper presents a high-resolution gridded dataset of daily minimum (TN), mean (TG) and maximum (TX) near-surface air temperatures over Poland, covering the period from 1951 to 2020, with a spatial resolution of 1 km². PL1GD-T

dataset was developed using radial basis functions (RBFs) applied to quality-controlled observations from ground weather stations from the Institute of Meteorology and Water Management – National Research Institute. Cross-validation methods evaluated the gridding procedure on a monthly basis. The linear RBF was employed by hold-out cross-validation (HO-CV) as the most suitable for the gridding procedure among other RBFs. The leave-one-out cross-validation (LOO-CV) was performed to ensure the ability to reproduce the original characteristics variability. The values of the scores averaged over all stations for

individual months are in range 0.3-0.2, 0.3-0.2, 0.1-0.2 for the bias, in range 1,23-1,46, 0.69-0.92, 0.84-0.99 the root-mean-squared difference (RMSD) and in range 0.91-0.97, 0.98-0.99, 0.98-0.99 for the correlation for TN, TG and TX, respectively. The RSMD is clearly altitude dependent, increasing from low-land to mountainous regions. The dataset's scope and resolution allowed robust estimation of local climate variability characteristics and observed trends. The availability of high-resolution datasets in both spatial and temporal contexts is essential for climate change impact analysis on a smaller scale. This new

dataset provides a quality-validated, high-resolution, and open-access dataset that could be utilised by society, administrative bodies, or research institutions for further analysis. The dataset is publicly available from the repository of the Institute of Meteorology and Water Management – National Research Institute under https://doi.org/10.26491/imgw_repo/PL1GD-T (Jaczewski et al., 2024).

## 1   Introduction

Contemporary climate change is one of the most significant issues facing civilisation. It affects societies in many ways, from direct physical impacts (droughts, floods, extreme events like tornadoes, heatwaves, and wildfires) to influencing the economies. We have no other way than to adapt, but our measures strongly depend on the quality and quantity of climate





information we can acquire. Applications to research, natural resource management, and infrastructure planning typically require adequate spatial and temporal resolution (often 30 years of climate normality) and a record of past climate change

(historical time series), you need long-term climate baseline data. The applicability of such data lies in the availability of measurement data from European National Meteorological and Hydrological Services (NMHS) networks and in the ability to assess climate change locally, which is more important in the context of adaptation. The spatial resolution of most NMHS station networks is limited, and in the case of Poland, it is c.a. 50 km$^2$ (depending on the relief). This is a severe impediment in the adaptation or mitigation analyses, preventing the proper assessment of climate variability. The research aimed to develop

a gridded database of daily mean, maximum and minimum temperatures (hereafter denoted as TG, TX and TN, respectively) with a spatial resolution of 1 km$^2$ (Jaczewski et al., 2024). The gridding procedures involved the application of RBFs (Radial Basis Functions) with several covariates, and the procedure's results were the subject of thorough comparison with in-situ measurements to ensure the ability to reproduce the original characteristics variability. The research outcome in a netCDF format database covers the 1951-2020 period, thus comprising 70 years of systematic measurements in the Polish NMHS

network. Such database scope and the daily temporal and 1 km$^2$ spatial resolution allowed robust estimation of local climate variability characteristics and observed trends analysis, which society, administrative bodies, or research institutions can further utilise.

Climate change impact analysis requires the availability of high-resolution data sets in both spatial and temporal contexts. This is especially crucial if the study is to be performed on a smaller scale (administrative, physical-geographic). This assumption

is rarely met with the stations' density operating in NMHSs. Thus, there is a clear need to prepare a quality validated dataset with open access in high temporal and spatial scales.

During the XXI century, there were several attempts to provide such data. The E-OBS gridded dataset is one of the most used European daily databases (https://www.ecad.eu/download/ensembles/download.php). However, the spatial resolution is relatively coarse (0.1° × 0.1° or 0.25° × 0.25°; c. a. 11 km$^2$ or 25 km$^2$, respectively), which in the latter case may be insufficient

for analyses of climatological conditions in small spatial scales. The present version is 28.0e and was updated in October 2023. However, a substantial advantage of this dataset is that it is derived from in-situ measurements. E-OBS is a daily gridded land-only observational dataset over Europe. The blended time series from the station network of the European Climate Assessment & Dataset (ECA&D) project forms the basis for the E-OBS gridded dataset. All station data are sourced directly from the NMHSs or other data-holding institutions. For many countries, the number of stations is the complete national network.

Therefore, it is much more dense than the station network routinely shared among NMHSs (which is the basis of other gridded datasets). The density of stations gradually increases through collaborations with NMHSs within European research contracts. The position of E-OBS is unique in Europe because of the relatively high spatial horizontal grid spacing, the daily resolution of the dataset, the provision of multiple variables and the length of the dataset. Finally, the station data on which E-OBS is based are available through the ECA&D webpages (where the data owner has permitted to do so). In these respects, it contrasts

with other datasets. Aside from E-OBS, other European databases were produced in the daily resolution. The set of gridded datasets covering Europe is provided in Table 1. Usually, the data available comprises mean and extreme daily temperature.



**Table 1.** Selected global and European air temperature gridded datasets with a high spatial and daily temporal resolution.

| Citation/dataset | Region | Variable | Interpolation method | Period | Spatial resolution |
|---|---|---|---|---|---|
| E-OBS v. 27.0e (Cornes et al., 2018) | Europe | TG, TN, TX | Monthly background fields from thin-plate splines and daily fields from the reduced-rank thin-plate spline | 1950-present | 0.1° × 0.1° |
| (Krähenmann et al., 2011) | Europe (WMO region 6) | TN, TX | Monthly background fields from block kriging and interpolation of daily anomalies from block kriging | 2005-2008 | 5 km × 5 km |
| (Ntegeka et al., 2014) | Europe | TG, TN, TX | Inverse distance weighting | 1990-2011 | 5 km × 5 km |

As seen in

Table 2, a vast range of methodologies is used in the calculations. Starting from the nearest neighbour (NN) approach through linear interpolation aided with IDW, distance weighting interpolation, thin plate splines, kriging, external drift kriging, inverse distance weighting, optimal interpolation, or combined methods. In most cases, the resolution rarely reaches 1 km$^2$ and is usually restricted to 5 km$^2$ or larger. In Central Europe, there are few daily databases of meteorological variables with a
resolution higher than 5 km$^2$. Such is available only for Germany, Austria, and Switzerland, but in all those cases, the temporal scope is relatively short, only for Switzerland (1961-2010) and Austria, which covers 60 years (1961-2013). This shows a need to produce gridded datasets of essential thermal characteristics on a daily scale covering the most extended possible period and utilising all possible, verified data quality station data. In the case of our research, we were able to extend the time scope back to 1951 and forward to 2020, which resulted in 70 years of daily data. Thus, for the reasons stemming from the needs of
possible end users and the scientific community, it is crucial to make an effort to provide such data for Poland.

Table 2. This table lists national gridded datasets containing temperature variables with at least daily temporal resolution.

| Dataset (citation) | Region | Variables | Interpolation method | Period | Temporal resolution | Spatial resolution | Validation method | Validation parameters |
|---|---|---|---|---|---|---|---|---|
| HYRAS (Razafimaharo et al., 2020) | Germany and its catchment areas | TN, TG, TX | interpolation method integrating non-linear elevation temperature profiles and IDW using multi-dimensional distances | 1951-2015 | daily | 5 km$^2$ | LOO-CV, Fivefold cross-validation, Comparison with other datasets | ME, MAE |
| TRY (Krähenmann et al., 2018) | Germany | T | According to (Frei, 2014; Hiebl and Frei, 2016) | 1995-2012 | hourly | 1 km$^2$ | LOO-CV | RMSE, q-q |
| (Frei, 2014) | Switzerland | TG | A spatial scale-separation approach | 1961-2010 | daily | 1 km$^2$ | LOO-CV | MAE, RMSE |
| SPARTACUS (Hiebl and Frei, 2016) | Austria | TN, TX | A spatial scale-separation approach | 1961-2013 | daily | 1 km$^2$ | LOO-CV | ME, MAE, RMSE |



| | | | | | | | | |
|---|---|---|---|---|---|---|---|---|
| FORBIO (Delvaux et al., 2015) | Belgium | TN, TX | EDK | 1980-2013 | daily | 4 km² | LOO-CV | RMSE |
| G2DC-PL+ (Piniewski et al., 2021) | Poland and its catchment areas | TN, TX | EDK | 1951-2013 | daily | 5 km² | LOO-CV | nRMSE |
| EMO-5 (Thiemig et al., 2021) | Europe | T, TN, TG, TX | IDW, modified SPHEREMAP, OK | 1990-2019 | daily 6-hourly | 5 km² | Comparison against a high-resolution regional grid | ME, MAE, MSE |
| STEAD (Serrano-Notivoli et al., 2019) | Spain | TN, TX | | 1901-2014 | daily | 5 km² | LOO-CV | r, MAE, ME, RM, RSD |
| seNorge_2018 (Lussana et al., 2019) | Norway | TN, TG, TX | A spatial scale-separation approach | 1957-2017 | daily | 1 km² | LOO-CV | MAE, RMSE |
| SLOCLIM (Škrk et al., 2021) | Slovenia | TN, TX | GLMMs, GLMs | 1950-2018 | daily | 1 km² | LOO-CV | r, MAE, ME, RM, RSD |
| seNorge2 (Lussana et al., 2018) | Norway | T, TN | a modification of classical OI | | daily, hourly | | | |

**Variables:** TN – daily minimum temperature, TG – mean daily temperature, TX – maximum daily temperature, T – sub-daily temperature

**Interpolation methods:** OK – ordinary kriging, OI – optimal interpolation, EDK - External Drift Kriging, IDW – Inverse Distance

Weighting, GLMM - generalised linear mixed models, GLM - generalised linear models

**Validation methods:** LOO-CV – the leave-one-out cross-validation

**Validation metrics:** r – correlation, (n)RMSE – (SD normalised) the root-mean-square error, RMSD – the root mean square difference, MAE – the mean absolute error, ME – the mean error (bias), RM – the ratio of means, RSD – the ratio of standard deviations, SD – standard deviation

## 2    Data and methods

### 2.1    Station data

Using a long series of meteorological data makes it possible to carry out analyses related to the impact of the variability of meteorological conditions over a selected area on various sectors of human activity.

Ensuring data quality is crucial in this case. IMGW-PIB has taken steps to develop complete information on the daily variability

of thermal and precipitation conditions in Poland since 1951. The analysis and prepared data series include information on the daily values of the following meteorological variables: average daily air temperature, maximum daily air temperature, and minimum daily air temperature.



The official method of calculating the characteristics, particularly the average daily air temperature, varied in different periods and depended on the station's measurement program, which was different at synoptic and climatological stations. One method

of calculating the average temperature was necessary to compare data and ensure homogeneity at the entire available range of stations in the whole period. Due to the use of data from both synoptic and climatological stations in the set, the same method was used to calculate the average daily air temperature:

$$TG = \frac{T_{06} + T_{18} + TX + TN}{4} \quad\quad (1)$$

This way, consistent information was obtained on the spatial variability of thermal conditions throughout the country in the

multiannual period. And even though in the case of synoptic stations, the values of the TG calculated in this way are not the official daily average (average of 8 measurements), the possibility of a uniform analysis throughout the country brings more benefits than potential losses resulting from a different calculation method. In this way, the so-called uniform air temperature series includes data for 347 stations from 1951 to 2020.

## 2.2   Spatial interpolation

RBFs have been successfully applied for interpolation station observational data in meteorology (Antal et al., 2021; Fathizad et al., 2017; Piri, 2017; Ryu et al., 2024; Saaban et al., 2013; Wang et al., 2014; Yang and Xing, 2021) or hydrology (Gao et al., 2022; Wypych and Ustrnul, 2011). RBF interpolation models have also been used in agricultural applications for prediction purposes (Rocha and Dias, 2018, 2019). RBF interpolation method could be defined by Eq. (2):

$$F(r) = \sum_{j=1}^{m} \alpha_j \varphi(r_j), \quad\quad (2)$$

Where $\alpha_j$ are the weights and $\varphi(r_j)$ RBF function of the distance $r_j$ between interpolation point to $m$ sample points.

Six RBFs were used and given as,

linear: $\varphi(r_j) = r_j$                                        (3)

cubic: $\varphi(r_j) = r_j^3$                                       (4)

thin plate spline: $\varphi(r_j) = r_j^2 \log r_j$                  (5)

gaussian: $\varphi(r_j) = e^{-\left(\frac{r_j}{\beta}\right)^2}$                 (6)

multiquadric: $\varphi(r_j) = \sqrt{1 + \left(\frac{r_j}{\beta}\right)^2}$        (7)

inverse multiquadric: $\varphi(r_j) = \frac{1}{\sqrt{1 + \left(\frac{r_j}{\beta}\right)^2}}$    (8)

where $\beta$ is shape parameter. In the three dimensional space the Euclidean distance is given by:



$$r_j = \sqrt{(x - x_j)^2 + (y - y_j)^2 + (z - z_j)^2} \tag{9}$$

For a given dataset of $m$ points, the interpolated $n$ values are obtained by following linear system of $n$ equations:

$$F(x_i, y_i, z_i) = \sum_{j=1}^{m} \alpha_j \varphi(r_j) \tag{10}$$

In our case, x and y correspond to geographical coordinates, and z is elevation. Generally, the coordinates are treated as predictors, and more explanatory variables can be introduced to the interpolation procedure. We intentionally limited the variables to geographical coordinates, as temperature and altitude depend most on them. Choosing the optimal value of the

shape parameter is not a straightforward task (Fasshauer and Zhang, 2007). Often, this problem is paved over in the application studies, and it should be assumed that the value has been chosen as the mean distance between the nearest points (observational stations) as it is implemented in numerical tools as the default option. We have used this value in the interpolation procedure.

### 2.3  Temporal and spatial reference of the dataset

The dataset consists of daily gridded TN, TG and TX covering Polish territory at 1 km$^2$ spatial resolution for 1951-2020. The

interpolation is in the grid projected in the PUWG 1992 (EPSG:2180) coordinate reference system. The elevation data originates from DTM for Poland (DTM 100m, 2023), corresponding to DEM. Given data, simplify the interpolation procedure as all three coordinates (x, y and z) are in the same units (meters). Additionally, we avoided topography interpolation by defining the grid as every tenth x and y coordinate of the 100 m × 100 m DTM mesh.

### 2.4  Validation

The most popular technique for validating interpolation techniques is leave-one-out cross-validation (LOO-CV). It involves using all except one observation data for interpolation, and the remaining station is used for validation. The main drawback of this method is relatively high computational time as the procedure has to be repeated for every station, which has to be removed from the whole dataset. Another cross-validation method, the hold-out technique (HO-CV), is used to avoid long calculations. This method splits the observation data into interpolation and validation sets. Daily, 95% of stations are randomly removed,

and 5% are used for validation. The procedure is repeated every day. The hold-out technique was chosen to select the RBF function, and the LOO-CV technique was used to evaluate results with the selected RBF function.

To evaluate the accuracy of the interpolated data mean error (ME), root mean square difference (RMSD), centred RMSD, and Pearson's correlation coefficient have been used. The two latter metrics and standard deviations have been presented through Taylor's diagrams (). The tables with numeric results rounded to two decimal points were provided, including differences in

the 5$^{th}$ and 95$^{th}$ percentiles. The observed data was compared to the nearest interpolation grid point, and validation was performed monthly. The following Equations define the metrics:

mean error (bias): $\mathbf{ME} = \frac{1}{N}\sum_{n=1}^{N}(\boldsymbol{I_n} - \boldsymbol{O_n})$ (11)





root mean square difference: $\mathbf{RMSD} = \sqrt{\frac{1}{N}\sum_{n=1}^{N}(I_n - O_n)^2}$ (12)

Centred root-mean-square difference: $\mathbf{cRMSD} = \sqrt{\frac{1}{N}\sum_{n=1}^{N}[(I_n - \bar{I}) - (O_n - \bar{O})]^2}$ (13)

Pearson's correlation coefficient: $\mathbf{r} = \frac{\frac{1}{N}\sum_{n=1}^{N}(I_n - \bar{I})(O_n - \bar{O})}{\sigma_I \sigma_O}$ (14)

Where $I$ and $O$ denote interpolated and observed values, N is the number of observations, and $\sigma_I$ and $\sigma_O$ are standard deviations of interpolated and observed values, respectively. $\bar{I}$ and $\bar{O}$ are the mean of interpolated and observed values, respectively.

## 3    Results

### 3.1    Hold-out cross-validation (HO-CV)

Using HO-CV, we will assess which RBF function is the most appropriate for interpolating temperature variables. This involves dividing the observation data into interpolation and validation sets. From the daily observational dataset, 95% of stations are randomly selected for interpolation, while 5% are used for validation. The process is repeated daily, resulting in varying interpolated and validation datasets. The validation process results are presented using Taylor diagrams, which provide a quick statistical summary of how well the modelled and observed data align. We show mean values of correlations, centred

root-mean-square differences, and standard deviations for all RBF methods every month. While a common approach is to normalise the differences and standard deviations of modelled data by the standard deviations of observations, it may negatively impact the clarity of the figures in our case. As the variability (SDs) of the variables differ across seasons and variables, we can present the results for all methods and temperatures in the exact figure.

The linear approach is undoubtedly the most accurate for all months and variables (Figure 1). Only for TX in April and August,

values interpolated by inverse multiquadric RBF function give similar agreement with observations. The mean cRMSD for linear RBF function is lower than 1 for TG and TX, but for TN, it exceeds 1. This method also sustains observed variability as the standard deviations of interpolated and observed values are similar. Correlations exceed 0.95 for all variables and months except TN from April to September. These results allowed us to proceed further only with the linear method.





**Figure 1.** Taylor diagrams for HO-CV of spatial interpolation of TN, TG and TX by different RBF functions.



## 3.2 Leave-one-out cross-validation (LOO-CV)

LOO-CV has emerged as a widely used verification methodology for spatial interpolation techniques in meteorology and climatology. In executing this method, the given station is removed from the daily observation dataset, and the remaining stations are used to construct the interpolated field. The value at the grid nearest to the removed station is compared to the observed value. The procedure is repeated for all stations and days in the analysed period, and the mean values of the comparison metrics are presented in Taylor diagrams (Figure 2). The charts depict the level of agreement between the observed and interpolated values. The results show that the best agreement is achieved for TG in winter and spring, with the lowest cRMSD and highest correlation values. Although the agreement for TX is slightly worse than the mean, it is still relatively favourable overall, with the best agreement seen in autumn. However, the interpolation methodology performs worse in capturing the nuances of TN, with the best agreement observed in winter. It is important to note that the variability of all temperatures is preserved in standard deviation, with similar values between observed and interpolated data. The precise values for these and additional metrics are in the following tables.





**Figure 2.** Taylor diagrams for LOO-CV of spatial interpolation of TN, TG and TX by linear RBF function.





Table 3 provides the set of quality metrics for RBF interpolation. Mean error (ME, a.k.a BIAS) is minimal and, for all variables, does not exceed 0.03 with minuscule interannual variability. TN is characterised by the highest RMSD reaching, on average, 1.29. The RMSD annual value for TG equals 0.75, whereas for the TX, it is slightly higher (0.92). The highest average annual value is recorded for TN (1.29). For all analysed variables, Pearson's correlation coefficient is very high. An annual course ranges from 0.94 (TN) to 0.98 (TG, TX). Only in the case of TN, there is a slight seasonal variability, with the lowest values

(0.91) in July and the highest (0.97) in winter (Dec-Jan). RSD (the ratio of standard deviation) is very high, nearing 1 for all variables with minimal seasonal variability. Such a situation suggests a slight variance flattening in the outcome RBF data and further assures the high quality of the RBF interpolation results.

The difference in the values of quantiles (5% and 95%) suggests slight underestimation in the case of Q95 and overestimation for Q05. This pattern repeats for all variables (TN, TG, and TX), with the highest differences in average annual values recorded

for TN.

**Table 3.** Overall monthly validation measures (OBS vs GRIDDED).

| VAR | | Jan | Feb | Mar | Apr | May | Jun | Jul | Aug | Sep | Oct | Nov | Dec |
|-----|-----|-----|-----|-----|-----|-----|-----|-----|-----|-----|-----|-----|-----|
| TN | ME | 0.03 | 0.03 | 0.03 | 0.03 | 0.03 | 0.03 | 0.03 | 0.03 | 0.03 | 0.03 | 0.02 | 0.02 |
| | RMSD | 1.46 | 1.45 | 1.31 | 1.40 | 1.44 | 1.38 | 1.37 | 1.41 | 1.39 | 1.38 | 1.23 | 1.32 |
| | cRMSD | 1.43 | 1.39 | 1.25 | 1.28 | 1.32 | 1.26 | 1.24 | 1.25 | 1.27 | 1.30 | 1.18 | 1.28 |
| | Pearson | 0.97 | 0.97 | 0.96 | 0.94 | 0.93 | 0.92 | 0.91 | 0.92 | 0.93 | 0.94 | 0.96 | 0.97 |
| | RSD | 0.99 | 0.98 | 0.98 | 0.97 | 0.96 | 0.96 | 0.95 | 0.96 | 0.96 | 0.97 | 0.98 | 0.98 |
| | Q95D | -0.12 | -0.06 | -0.18 | -0.17 | -0.15 | -0.14 | -0.17 | -0.22 | -0.23 | -0.13 | -0.18 | -0.13 |
| | Q05D | 0.27 | 0.27 | 0.25 | 0.25 | 0.31 | 0.28 | 0.33 | 0.32 | 0.35 | 0.23 | 0.14 | 0.28 |
| TG | ME | 0.03 | 0.02 | 0.02 | 0.02 | 0.02 | 0.02 | 0.02 | 0.02 | 0.02 | 0.02 | 0.02 | 0.02 |
| | RMSD | 0.92 | 0.87 | 0.77 | 0.78 | 0.74 | 0.70 | 0.69 | 0.74 | 0.78 | 0.84 | 0.78 | 0.85 |
| | cRMSD | 0.94 | 0.85 | 0.74 | 0.72 | 0.68 | 0.64 | 0.62 | 0.67 | 0.71 | 0.80 | 0.76 | 0.86 |
| | Pearson | 0.99 | 0.99 | 0.99 | 0.98 | 0.98 | 0.98 | 0.98 | 0.98 | 0.98 | 0.98 | 0.98 | 0.98 |
| | RSD | 0.99 | 0.99 | 0.99 | 0.99 | 0.99 | 0.98 | 0.99 | 0.99 | 0.99 | 0.99 | 0.99 | 0.99 |
| | Q95D | -0.10 | -0.02 | -0.04 | -0.02 | -0.04 | -0.07 | -0.07 | -0.10 | -0.02 | -0.02 | -0.11 | -0.03 |
| | Q05D | 0.20 | 0.14 | 0.14 | 0.10 | 0.06 | 0.04 | 0.10 | 0.12 | 0.06 | 0.16 | 0.08 | 0.07 |
| TX | ME | 0.03 | 0.02 | 0.02 | 0.02 | 0.02 | 0.01 | 0.01 | 0.01 | 0.01 | 0.02 | 0.02 | 0.02 |
| | RMSD | 0.99 | 0.94 | 0.91 | 0.95 | 0.98 | 0.96 | 0.92 | 0.90 | 0.84 | 0.89 | 0.91 | 0.95 |
| | cRMSD | 1.02 | 0.93 | 0.90 | 0.96 | 0.95 | 0.94 | 0.87 | 0.87 | 0.82 | 0.89 | 0.92 | 0.97 |
| | Pearson | 0.98 | 0.98 | 0.99 | 0.98 | 0.98 | 0.98 | 0.98 | 0.98 | 0.98 | 0.98 | 0.98 | 0.98 |
| | RSD | 0.99 | 0.99 | 0.99 | 0.99 | 0.99 | 0.99 | 0.99 | 0.99 | 0.99 | 0.99 | 0.99 | 0.99 |
| | Q95D | -0.07 | -0.06 | -0.09 | -0.05 | -0.06 | -0.11 | -0.05 | -0.08 | -0.11 | -0.14 | -0.07 | -0.11 |
| | Q05D | 0.16 | 0.15 | 0.12 | 0.13 | 0.22 | 0.13 | 0.11 | 0.11 | 0.08 | 0.16 | 0.08 | 0.13 |

Meteorological networks in mountainous regions across many countries are often sparse. Additionally, the station density in

Poland is relatively low compared to other European countries. Moreover, considering the relatively high climate gradients and heterogeneous and rapidly changing weather-influenced mountainous areas, they are worse represented in gridded fields. Verification of RBF interpolation quality based on RMSD for distinct altitude (m a. s. l.) classes confirms overall quality characteristics, but there is an apparent increase in RMSD values with the rise in altitude for TN. Up to 300 m, the RMSD



values are relatively uniform throughout the year and range from 1.08 in November to 1.35 in May. On average, the annual
RMSD value equals 1.26. In the 300-500 m altitude class, there is an increase in RMSD value to 1.58 (annual) with the range
from 1.45 (June) to 1.83 (January). The next class (500-1000 m) shows yet another increase to 2.20 (annual average), with the
range between 2.00 (June) to 2.65 (January). In the highest class above 1000 m, the RMSD value increases again, reaching an
average of 2.59 with a range between 2.24 (June, July) and 3.16 (January). What is apparent is the increase in seasonal RMSD
variability with altitude, with relatively small interannual differences at a lower altitude to the greatest in the case of the highest
altitude class.

For daily mean air temperature (TG), the overall pattern of RMSD increase with altitude is confirmed. However, what is
apparent is that this RMSD increase is much smaller than in the case of extreme temperatures. The average annual values range
from 0.71 (0-300m) to 1.97 (> 1000 m). The seasonal variability in the lowest altitude class is minimal and ranges from 0.65
(July) to 0.77 (January). With the altitude increase, the seasonal variability increases with higher RMSD values in colder
months and lower during the warm part of the year. At 300-500 m, the annual average RMSD increases slightly to 0.91, but
what appears is more apparent seasonal variability with values ranging from 0.72 (June) to 1.18 (January). For the following
two altitude classes, the average annual RMSD exceeds 1.00. In the 500-1000 altitude class, the annual average RMSD equals
1.33; stations above 1000 m reach 1.97. The increased seasonal variability with the RMSD range for both classes at 0.96 is
also typical for higher altitude types.

TX follows the pattern mentioned above with some distinctive differences. The annual average values range from 0.88 (0-300
m) to 2.14 (> 1000 m). What strikes relatively small RMSD values exist for two lower altitude classes closer to the TG RMSD
values than TN. Also, the seasonal variability is less pronounced than for TN. The highest interannual variability of TX RMSD
is recorded for the 500-1000 m altitude class, ranging from 1.00 (July) to 1.82 January).

**Table 4.** Overall monthly RMSD (OBS vs. GRIDDED) values for station altitude classes.

|     | altitude class | **Jan** | **Feb** | **Mar** | **Apr** | **May** | **Jun** | **Jul** | **Aug** | **Sep** | **Oct** | **Nov** | **Dec** |
|-----|----------------|------|------|------|------|------|------|------|------|------|------|------|------|
| **TN** | 0 - 300    | 1.28 | 1.30 | 1.20 | 1.29 | 1.35 | 1.32 | 1.29 | 1.32 | 1.31 | 1.25 | 1.08 | 1.14 |
|     | 300 - 500      | 1.83 | 1.72 | 1.51 | 1.56 | 1.56 | 1.45 | 1.47 | 1.53 | 1.55 | 1.62 | 1.49 | 1.65 |
|     | 500 - 1000     | 2.65 | 2.47 | 2.05 | 2.20 | 2.19 | 2.00 | 2.03 | 2.17 | 2.12 | 2.13 | 2.00 | 2.37 |
|     | >1000          | 3.16 | 2.88 | 2.56 | 2.43 | 2.38 | 2.24 | 2.24 | 2.37 | 2.49 | 2.85 | 2.60 | 2.85 |
| **TG** | 0 - 300    | 0.77 | 0.76 | 0.70 | 0.72 | 0.71 | 0.68 | 0.65 | 0.69 | 0.71 | 0.74 | 0.66 | 0.70 |
|     | 300 - 500      | 1.18 | 1.04 | 0.88 | 0.86 | 0.79 | 0.72 | 0.73 | 0.80 | 0.87 | 0.97 | 0.95 | 1.08 |
|     | 500 - 1000     | 1.90 | 1.62 | 1.27 | 1.24 | 1.09 | 0.94 | 0.97 | 1.11 | 1.24 | 1.40 | 1.43 | 1.74 |
|     | >1000          | 2.52 | 2.18 | 1.88 | 1.83 | 1.77 | 1.56 | 1.57 | 1.67 | 1.85 | 2.22 | 2.20 | 2.38 |
| **TX** | 0 - 300    | 0.85 | 0.84 | 0.87 | 0.99 | 1.03 | 0.99 | 0.92 | 0.88 | 0.79 | 0.80 | 0.79 | 0.81 |
|     | 300 - 500      | 1.23 | 1.10 | 0.98 | 0.95 | 0.94 | 0.93 | 0.90 | 0.91 | 0.91 | 1.02 | 1.10 | 1.18 |
|     | 500 - 1000     | 1.82 | 1.51 | 1.23 | 1.11 | 1.07 | 1.03 | 1.00 | 1.03 | 1.09 | 1.29 | 1.51 | 1.77 |
|     | >1000          | 2.51 | 2.25 | 2.06 | 2.15 | 2.11 | 1.92 | 1.84 | 1.85 | 1.93 | 2.20 | 2.38 | 2.47 |



Figures Figure 3, Figure 4, Figure 5 present the spatial variability of RMSD for all analysed variables on a monthly scale (with seasons in rows beginning from spring). There is an apparent pattern with relatively low RMSD values in central Poland in the lowlands and substantially higher in the mountains. This pattern is quite common for other variables in most months.

Suppose one divides RMSD values into bins of equal width, equalling 1 with breaks from 0 to 5. In that case, one can assess the RBF method's overall quality and share the station's occurrence in those five bins, with the first referring to the 0-1 RMSD boundaries and the fifth with values of RMSD between 4 and 5.

TN (Figure 3) exhibits the most significant overall variability, and it is the only variable for which the fifth bin (RMSD values from 4 to 5) is filled in the annual cycle. However, the fifth bin's average annual share of stations is only 0.1% (3 cases out of

4164). Also, there is a noticeable difference between the share of the first bin, which, on average, is only 57.8% of the cases. In contrast, for TG and TX, it is over 90%, which suggests that in the case of minimum temperature, the RBF interpolations' general quality seems lower than in the case of the other two variables. The second bin (RMSD - 1 to 2) comprises, on average, 39.7% of cases, which is substantially higher than for TG and TX. In the case of the first two bins, there is also high seasonal variability. For the first bin, the values range from 51.3% (May) to 70.9% (November). For the second bin (RMSD values

from 1 to 2), the share varies from 27.1% (November) to 45.8% (May).

TG (Figure 4), an even higher share (94.8%) of the stations falls into the first RMSD bin (0-1 values). The seasonal variability is relatively low, ranging from 89.3% (January) to 98.0% (June, July). In the second bin (on average 4.4% of the stations), there is a substantial seasonal variability with RMSD values ranging from 1.4% (June) to 8.9% (January, February)

The overall share of RMSD values in the two upper bins (third and fourth) is less than 1%, with more pronounced seasonal

variability in the third bin.

TX (Figure 5), on average (annual), over 90% of stations fall into the first bin (0 to 1 RMSD value). The seasonal variability is relatively low and ranges from 86.5% (January) to 94.5 (September). The second bin's (1-2 RMSD value) average equals 8.5%, with more pronounced seasonal variability ranging from 4.9% (September, October)) to 12.1% (January). Two higher classes of RMSD values (2-3 and 3-4) comprise, in total, on average, only 0.9% of stations, but in the case of the 2-3 class,

one can see a high seasonal variability with RMSD values ranging from 0.3 to 1.4 (May).

TMIN - RMSD

**Figure 3**. Monthly RMSD values (degrees Celsius) for minimum daily temperature (TN) at selected stations.







Figure 4. Monthly RMSD values (degrees Celsius) for mean daily temperature (TG) at selected stations.








TMAX - RMSD

**Figure 5.** Monthly RMSD values (degrees Celsius) for maximum daily temperature (TX) at selected stations.




Derived with RBF data series may be a source of valuable information on recorded climate change characteristics in Poland. The daily temporal resolution of the data set allows the possibility of deriving multiple statistical characteristics with very high spatial resolution and detailed analysis of the thermal characteristics on the regional or even local scale (counties, small cities, or villages). For example, Figure 6 provides insight into the spatial variability of TX 90% quantile in Poland for two 30 years:

1951-1980 and 1991-2020 in January and July. As expected in January, there is a substantial shift in the values of TX. The overall range of recorded values changes from –1.4°C - 7.7°C (1951-1980) to 1.1°C-10.6°C (1991-2020), which indicates a 2.5°C shift in the lower boundary and 2.9°C in the upper indicating higher variability in the maximum values of TX. The overall spatial pattern of the TX 90% quantile remains relatively intact. However, the shift is visible in the whole country. In July, the change in the range of TX 90% quantile also reaches 2°C from 13.6°C-30.8°C to 15.6°C-32.9°C. This is also clearly

visible for nearly the whole country but less pronounced in the mountains and the vicinity of the Baltic Sea.

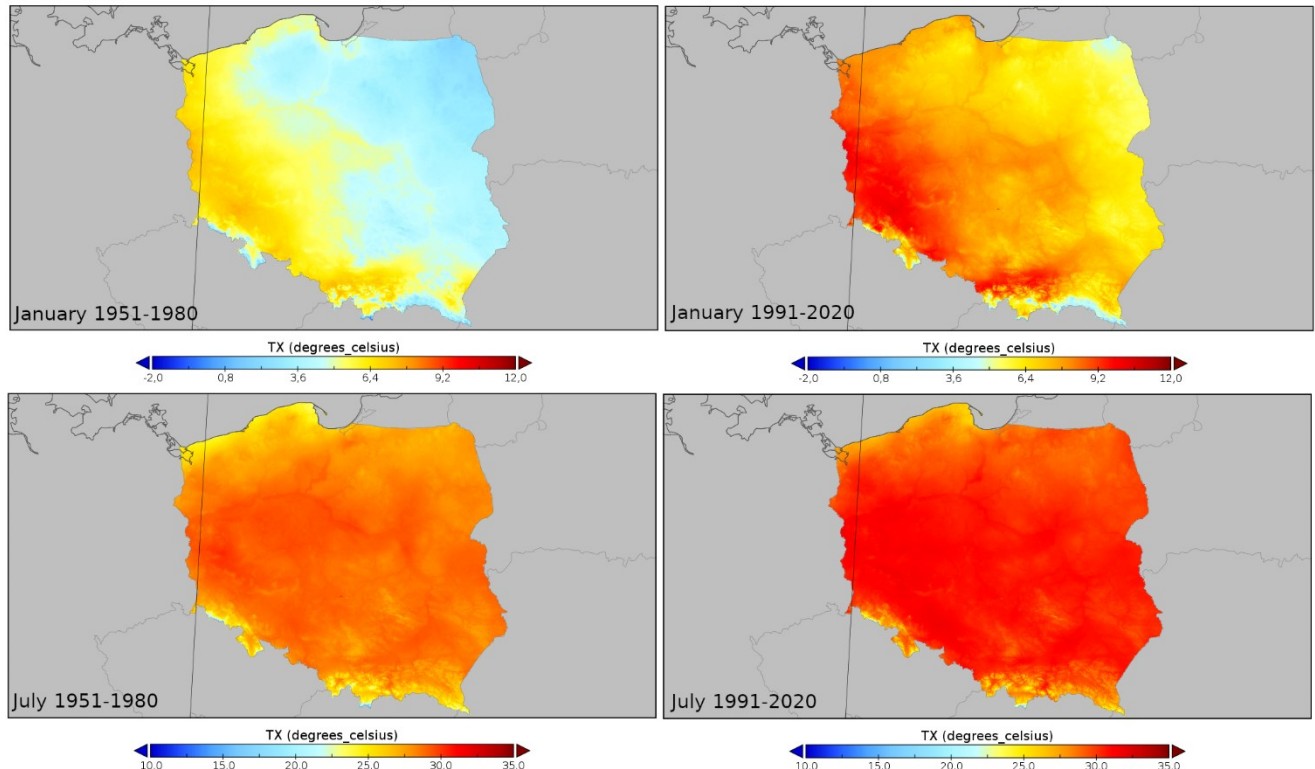

**Figure 6.** Spatial variability of 90% quantile of TX in Poland for selected 30-year periods (1951-1980 left panel, 1991-2020 right panel) in January (upper panel) and July (lower panel).





Figure 7 presents a similar analysis, but in this case, it is for the minimum temperature (TN) probabilistic characteristics. Again, the comparison between selected 30-year periods shows a substantial change in the derived values, concordant with the observed trends in the TN values in Central Europe (Ustrnul et al., 2021). For January, the change in the range of recorded values is 1.3C (an increase from –21.1°C to –19.8°C) in the case of minimum values and 1.2°C (an increase from –8.1°C to –6.9°C) for maximum grid values. Also, in this case, the overall observed change is characteristic for nearly the entire country, except the Baltic Sea coast, where the positive change is less pronounced. In July, the overall change in the range of values is +2.2°C (for minimum) and +0.9°C (for maximum values).

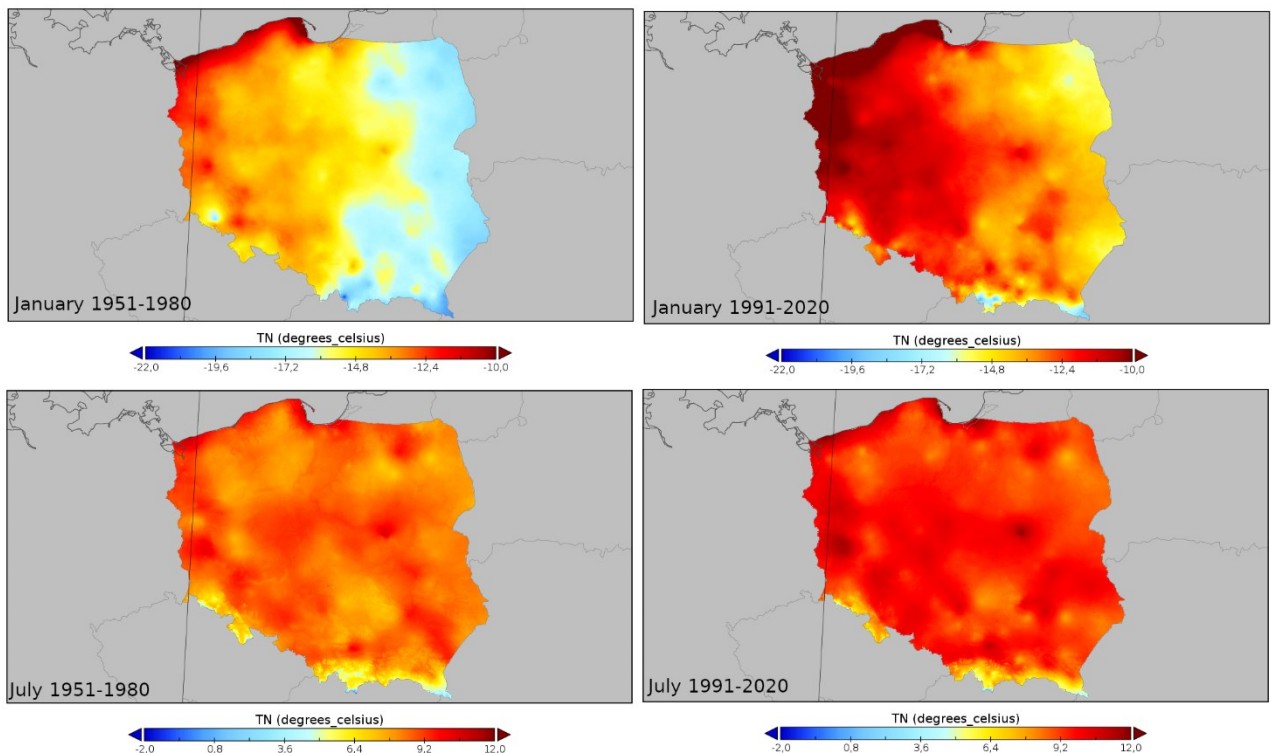


**Figure 7. Spatial variability of 10% quantile of TN in Poland for selected 30-year periods (1951-1980 left panel, 1991-2020 right panel) in January (upper panel) and July (lower panel)**

Figure 8 and Table 5 provide an additional example of the potential application of the data set. In this case, we present it for the whole country. Still, with the spatial resolution of ca.1km, it is equally applicable for the counties or even smaller areas desired by, for example, organisations preparing climate adaptation plans. The analysis presents the differences between the values of selected characteristics (in this case, extremes: 90% quantile of TX and 10% quantile of TN) for selected 30-year

periods: 2020-1991 and 1980-1951. The map indicates the areas where the observed climate change (expressed as the difference in temperature's extreme characteristics) was most pronounced and also allows further insight into the spatial

variability of chosen metrics.

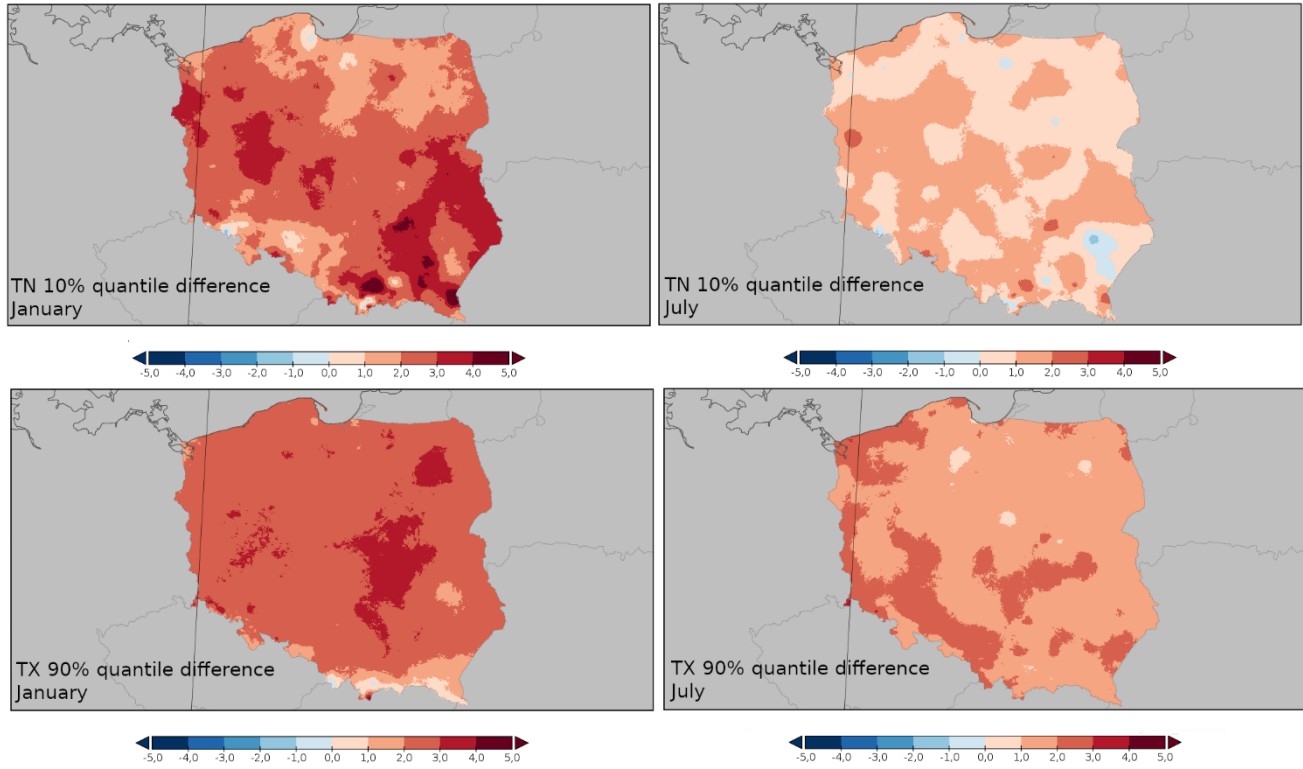

**Figure 8.** January (left) and July (right) differences (°C) between the values of TN 10% quantile (upper panel) and TX 90% quantile (lower panel) between periods 1991-2020 and 1980- 1951


The further analysis of such fields provides additional context. Table 5 shows the precise information on the area average change in the values of the abovementioned characteristics. It not only refers to the averages but, thanks to the calculation of the percentiles, provides information on the distribution of the shift magnitude in a given area. In this case, providing the percentiles' values cutting off the extreme 10% per cent of grid point values. For example, it shows that the area average change

for TX and TN quantiles (10% and 90%, respectively) average was relatively similar in January (ca. +2.5°C) and much lower in July; however, in July, this change was more pronounced for TX (+1,75°C) than TN (+0.96°C) extremes.

In the case of the TX 90% quantile, 10% of the analysed area experienced a shift of over +3.00 °C in January. For June, this value is lower and equals +2.15°C. For the TN 10% quantile, 10% of the area had a shift more significant than +3.28°C in January, whereas in July, it was much lower (+1.44°C).





The lower quantile (10%) of the TX 90% and TN 10% quantiles, showing the threshold for the area experiencing less or equal values, also indicates a more pronounced shift magnitude in January than in July. It is worth noting that in the case of TX, 90% quantile in January, only 10% of the country experienced a change lower than +2.13°C (1.31°C in July). In the case of TN 10% quantile, this is only +0.46°C in July and +1.65°C in January.

**Table 5.** Essential field characteristics (mean, quantiles: 0.1, 0.5 and 0.9) of TX90% and TN10% quantile differences (°C) between 1951-1980 and 1991-2020.

| Variable | Month | Mean | Q10 | Q50 | Q90 |
|---|---|---|---|---|---|
| TX 90% | Jan | 2.58 | 2.13 | 2.66 | 3.00 |
| | Jul | 1.75 | 1.31 | 1.76 | 2.15 |
| TN 10% | Jan | 2.45 | 1.65 | 2.45 | 3.28 |
| | Jul | 0.96 | 0.46 | 0.97 | 1.44 |

## 4 Example application of the datasets

One of the most important features required from grid data, which are to be used in climatological analysis and subsequently in the application of climate adaptation schemes, is the ability to represent the trends of analysed variables. Figure 9 presents
the inter-annual course of trend coefficients for TAVE, TX and TN at selected stations. The locations represent the low-lying stations (Hel, Łódź-Lublinek, Wrocław) and stations from mountainous areas: Zakopane (~900 m a.s.l.) and Kasprowy Wierch (1984 m a.s.l.). There is striking complete coherence of trend coefficients for stations outside the mountainous areas, where the values are virtually equal when comparing observational (OBS) and interpolated (RBF) data. More significant discrepancies occur in the mountainous regions, but the differences are within the 0.05°C/10 years range. In the case of
Zakopane, there is a systematic difference in the case of minimum temperature (TN) with slightly higher trend coefficient values for observational data. This situation, but to a much lower extent, is also visible in mean temperature (TAVE). In the case of Kasprowy Wierch (a station which is a mountain observatory and located in the terrain of highly variable relief), the differences in trend coefficients present themselves in the case of all analysed variables. For TX, there is a bias with higher trend values for interpolated values most visible in cold months (with the difference not exceeding 0.05°C/10 years, however).
It is less evident for TAVE, where greater bias is recorded for winter/cold months (October – March), with much more minor differences for the warm half of the year. Minimum temperature exhibits a reversing pattern with higher values of observed trends in the warmer part of the year and reversed bias (RBF > OBS) during cold months (October-March). The above examples confirm the possibility of using the RBF gridded data to analyse multiannual air temperature variability in Poland, including the areas of high relief variability.

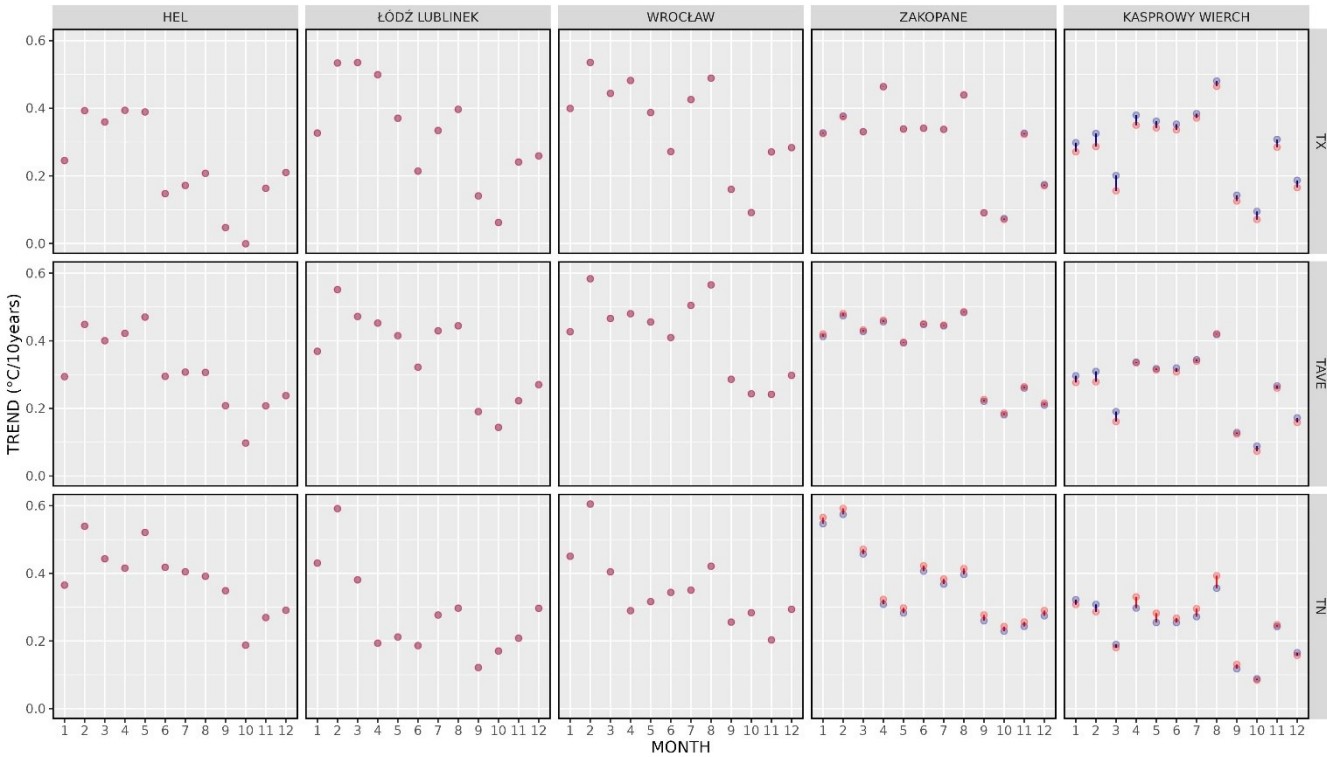


**Figure 9.** Monthly trend coefficients (°C/10 years) at selected stations 1951-2020. Red dots – OBS, blue dots – RBF

## 5    Summary

The paper presents a comprehensive study on developing a high-resolution gridded dataset of daily air temperatures at 2 m

above the ground (minimum, mean, and maximum) over Poland from 1951 to 2020, with a spatial resolution of 1 km². The dataset was created using quality-controlled observations from ground weather stations. Hold-out cross-validation was chosen to select appropriate radial basis functions (RBFs) for interpolation. Leave-one-out cross-validation assessed product quality on a seasonal, monthly, and station basis. The findings indicate the best performance for mean temperature in winter and spring, with high correlation and low root mean square deviation. Although the performance for maximum temperature is

slightly below average, it still shows relatively good agreement, especially in autumn. The method is less successful at accurately capturing minimum temperature, with the best results seen in winter. The analysis also confirms that the standard deviation of temperatures, representing variability, is consistent between observed and interpolated data. All variables have a bias that does not exceed 0.03, with minimal interannual variability. Pearson's correlation coefficient is very high, ranging from 0.94 to 0.98. The difference between the 5th and 95th percentiles suggests slight underestimation for Q95 and

overestimation for Q05. There is a noticeable increase in seasonal RMSD variability with altitude, with relatively small interannual differences at a lower altitude to the greatest in the case of the highest altitude class. Finally, we will show an example of the application of the resulting gridded product in the field of climate change.

This open-access dataset is crucial for climate change impact studies on a smaller scale and can serve a wide range of users, including researchers, administrative bodies, and society. One important application of such dataset is serving as reference

data for bias correction of regional dynamical downscaling results (e.g., EURO-CORDEX initiative) to elaborate effective adaptation and mitigation strategies.

## 6    Data availability

PL1GD-T - gridded dataset of the mean, minimum and maximum daily air temperature at the level of 2 m for the area of Poland at a resolution of 1 km × 1 km – are publicly available in the public repository of the Institute of Meteorology and

Water Management – National Research Institute under the https://doi.org/10.26491/imgw_repo/PL1GD-T (Jaczewski et al., 2024). The data are archived as NetCDF files in 10-year chunks for every variable separately. The resulting datasets consist of fields with values rounded to one decimal with data values encoded as integers with a scale factor of 0.1. It allowed the files' size to be decreased by a factor of two. The naming convention is *variable_startyear_endyear.nc* with *variable* prefixes 'tn', 'tg', and 'tx' for minimum, mean and maximum temperature, respectively. The NetCDF files comply with the CF-1.6

convention.

*Author contribution*. AJ designed the study with conceptual support from MMi and MMa. AJ developed the model code and prepared the dataset for the repository. AJ and MMa performed calculations for model validation. MMa prepared spatial maps. AJ and MMa prepared the manuscript. MMi contributed with critical review and commentary.


*Competing interests*. The authors declare that they have no conflict of interest.

*Acknowledgements*. The study was financially supported by the Institute of Meteorology and Water Management – National Research Institute under the project FBW10/2023, "Implementation of advanced monitoring and analysis of contemporary

climate variability in Poland in the context of observed climate change on a regional scale". Computations were supported using the computers of Centre of Informatics Tricity Academic Supercomputer & Network.



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
