# Peer review of "PL1GD-T: A high-resolution gridded daily air temperature dataset for Poland"

_Earth System Science Data, 2024_

## Referee Comment (RC1)

This manuscript documents and validates a daily gridded temperature dataset for Poland, designed for climatological applications. While the methodology is not innovative, it is appropriate for the authors' goals. The validation process uses recommended best practices and metrics, with the manuscript's primary strength being the detailed description of this validation. However, the communication of the research could be improved. With revisions addressing the following comments, the manuscript could be suitable for publication.

Main comments:

1. Station Data Description:
The section on station data requires significant expansion. Specifically:
    a)  Are the data homogenized?
    b)  Do you use a fixed number of stations over time, or do you include all available stations, potentially varying with time (as suggested in line 73)?
    c)  If station number varies over time, how does this impact the dataset's time consistency?
Points a) and b) should be stated in the Abstract too.

2. Temperature Definitions:
    a)  Clearly define TG, TX, and TN.
    b)  Specify the day definition used (e.g., 00 UTC to 24 UTC).
    c)  Does TG share the same day definition of TX and TN?

3. Methods and Equations:
    a)  Define all symbols in equations (e.g., "r" in Eq. (2)).
    b)  State fixed values, such as the number of sample points ("m").
    c)  Provide parameter values for the methods used.
    d)  Clarify the statement in lines 123–124: Are you not using elevation as a predictor? This needs to be explicit.

4. Manuscript Organization. The structure of the manuscript should be revised, some sections need to be moved and you should check for repetitions in the text. Some suggestions follows:
    a)  Abstract: Avoid unclear phrases like ""The linear RBF was employed by hold-out cross-validation (HO-CV) as the most suitable for the gridding procedure among other RBFs." If HO-CV was used for parameter optimization, clarify this. Listing specific scores may not be necessary in the abstract.
    b)  Introduction: Clearly state motivations, research questions, and approach. Avoid mixing conclusions into this section (e.g., lines 36–42). Eliminate redundancies (e.g., lines 43–45). Clarify if line 60 refers exclusively to observational datasets.
    c)  Reorganization: Consider moving Section 2.4 to the beginning of Section 3, avoiding redundancies (e.g., lines 156–157).
    d)  Section 4: Reassess why trend analysis is treated as an application while extremes analysis in Section 3 is treated as validation. Both seem to be dataset applications. Provide a clear strategy for distinguishing these analyses.

5. Dataset Purpose: State whether this dataset is intended for near-real-time climate monitoring (e.g., with regular updates) or as a one-time release

6. Title. A more concise title, such as "PL1GD-T: High-Resolution Gridded Daily Air Temperature Dataset for Poland," is recommended.

Minor comments:

Ensure consistent use of terms for TG, TN, and TX (e.g., TG is inconsistently referred to as TMEAN in Fig. 4 and TAVE in Section 4). Standardize abbreviations throughout the manuscript: e.g. replace 90% quantile and 10% quantile with 90th percentile and 10th percentile.

Line 260: This sentence seems abrupt and lacks context, making it feel disconnected from the surrounding text. Consider revising it to provide a smoother transition and explain its relevance to the discussion.

---

## Referee Comment (RC2)

**Review of essd-2024-433**

**PL1GD-T - gridded dataset of the mean, minimum and maximum daily air temperature at the level of 2 m for the area of Poland at a resolution of 1 km × 1 km**

The authors develop a gridded dataset of daily temperatures for Poland covering the 1951-2020 period, based on observed data from 347 stations. The research is oriented to the validation of the dataset. A very brief example of use is shown at the end of the manuscript.

While the overall objective of constructing a new daily dataset of temperatures for Poland is justified and well contextualized in the introduction, there are several unclear parts regarding the utilized data and the methodological approach.

For instance, regarding the **original data from stations**, a complete characterization of the raw temperature dataset is needed: Where are located the stations? (maybe a map would be useful); Do all of them have daily data or they come from hourly information? Are they automatic or manual stations? How many years of data (and gaps) they have? How is the temporal evolution of data availability? Did you apply any quality control/homogenization/gap filling procedure? This information and a basic statistical characterization are basic to contextualize the starting point of the grid and to understand the results.

In order to be clearer in the **methodological section**, the use of RBF must be further justified with more than a few references (Lines 105-108), mostly not related to temperature. While RBF is not particularly wrong, it is known as a "conservative" interpolation procedure, meaning that it usually reduces the spatial variability of the results which could be a problem in complex orography areas. As an example, the noted underestimation of higher values and overestimation of lower ones, can be attributed, with high confidence, to the interpolation scheme. Same situation can be observed at high elevations. Although higher RMSD values can be related to the scarcity of observations at those altitudes, the RBF is also probably related. None of the cited and well-known datasets used this approximation. Although it can be perfectly valid for this dataset, it must be supported by a justification. In addition, a gridded dataset must include a measure of uncertainty for all estimates to evaluate the reliability of the data at each location and time step.

Lastly, there is not a discussion section in which the new gridded dataset could be compared to others covering the region and cited in the introduction.

Apart from a general recommendation of a professional proofreading of English in the document, due to some confusing expressions, here are the minor comments, line by line:

**Introduction:**

L31: This statement is not valid for non-European regions.

L36: Which is the rationale to choose this spatial resolution?

L71: Actually, there is an operational product starting in 1960 that is updated regularly: https://www.meteoswiss.admin.ch/dam/jcr:818a4d17-cb0c-4e8b-92c6-1a1bdf5348b7/ProdDoc_TabsD.pdf

**Data and methods:**

L89: "IMGW-PIB". What is this?

L101-103: This means that all of the stations had hourly data?

L120: How many "m" points you used? Did you set a radius of search? This is important since the availability of stations is not the same throughput the temporal period, and it has an impact on the estimation.

L132-133: If I understood, you're assigning the value of a small 100x100m elevation pixel to the 100 pixels overlapped by a 1km$^2$, right? The problem with this is that is in those areas with high elevation variability you are assigning a non-representative value to the larger pixel and that can lead to significant biases in temperature estimates. A correct approach would be using the mean or the median elevation of all 100x100 pixels overlapped by one 1x1km pixel.

L139-141: While this is valid to evaluate daily estimates, it cannot be used to evaluate long term trends or even monthly or annual aggregates because, for the comparison between a single pixel and their overlapping observations, you have non-continuous data or even data from different stations. To avoid this issue, it is usual to separate some complete data series (for example 20-30% of the total) and use them to validate the estimates at those locations. In addition, how did you randomly select the 5%? It should be a spatially driven randomization to avoid spatial biases in selecting stations. This procedure is implemented in almost all GIS software.

**Results and discussion:**

L155-158: This is already stated in previous section

L170 (Figure 1): What is the meaning of coloured lines? Please extend the figure caption.

L193-195: Any interpretation for this? I guess that the RBF is smoothing the extremes.

L196 (Table 3): What is "cRMSD"? About Q95D and Q05D, I guess that they are the difference between observed and estimated Q95 and Q05, respectively. Please extend the table caption to make clear the meaning of all acronyms and the units in which the values are expressed.

L225 (Table 4): At this point, the ratio of means or the ratio of standard deviations would be a better test, since the RMSD is an absolute value and we can't see here if there is a bias related to an over- or under-estimation by altitudes.

Figures 3, 4 and 5: The tonal variation in a single-color scale avoids a correct interpretation. I recommend using a sequential color scale.

L261: Just a conceptual note: you can derive past temperature variability, but this is not (or not necessarily) related to climate change.

L267: These values are surprisingly high. Please clarify if you are showing the average 90th percentile of TX for both periods or the absolute maximum one.

L290-291: This is not results and it should be removed.

L315 (Table 5): Please, state if these values are the average of all the pixels.

**Example of application:**

L320: Trends significance is not shown. In addition, when you say "selected stations", do you mean the pixels overlapping those cities?

L322-323: This is not a fair comparison since you used the data from stations to build de gridded dataset.

---

## Author Comment (AC1)

**RC1: 'Comment on essd-2024-433', Anonymous Referee #1, 01 Dec 2024**

Dear referee, thank you for your insightful comments and suggestions, which have significantly improved the clarity and rigour of our manuscript according to your review.

We have thoroughly revised our manuscript and addressed all the questions and concerns raised by the reviewer. Detailed responses are provided in blue following the reviewer's comments (in black).

This manuscript documents and validates a daily gridded temperature dataset for Poland, designed for climatological applications. While the methodology is not innovative, it is appropriate for the authors' goals. The validation process uses recommended best practices and metrics, with the manuscript's primary strength being the detailed description of this validation. However, the communication of the research could be improved. With revisions addressing the following comments, the manuscript could be suitable for publication.

Based on the reviewers' comments, we have extended the Summary section with an outlook part on planned improvements in subsequent versions of the dataset within the presented methodology.

Main comments:

1. Station Data Description:

The section on station data requires significant expansion. Specifically:

a) Are the data homogenized?

The homogenisation was not included in our methodology. Our approach focuses on preserving the same consistent way of calculating the daily average temperature. We have extended the station data section to comment on this issue and describe the part on data QC in more detail.

b) Do you use a fixed number of stations over time, or do you include all available stations, potentially varying with time (as suggested in line 73)?

Thank you for your comment. We have conveyed it at the end of section 2.1. A figure showing daily data availability (Figure 2a) has also been added and commented on.

c) If station number varies over time, how does this impact the dataset's time consistency?

To consider this issue, we showed the distribution of available data percentage in the period in Figure 2b. It is commented on at the end of section 2.1.

Points a) and b) should be stated in the Abstract too.

We have included additional information on b) in the abstract. We decided not to extend the description of the source data according to a) remark. The reader is supposed to get familiar with the data description in a dedicated section.

2. Temperature Definitions:

a) Clearly define TG, TX, and TN.

The definitions are added to the description of Eq. (1)

b) Specify the day definition used (e.g., 00 UTC to 24 UTC).

The definition is added to the description of Eq. (1).

c) Does TG share the same day definition of TX and TN?

TX and TN are defined the same way throughout the period, and TG is calculated consistently. The description is improved to describe the methodology (Section 2.1) clearly.

3. Methods and Equations:

a) Define all symbols in equations (e.g., "r" in Eq. (2)).

We have corrected this part by improving the description of equations. (2), (9), (10).

b) State fixed values, such as the number of sample points ("m").

We have corrected it by describing how $m$ is taken into the interpolation procedure.

c) Provide parameter values for the methods used.

We have corrected it by providing the value of the shape parameter.

d) Clarify the statement in lines 123–124: Are you not using elevation as a predictor? This needs to be explicit.

Yes, elevation is a predictor. We have clarified it in the last paragraph of section 2.2.

4. Manuscript Organization. The structure of the manuscript should be revised, some sections need to be moved and you should check for repetitions in the text. Some suggestions follows:

a) Abstract: Avoid unclear phrases like ""The linear RBF was employed by hold-out cross-validation (HO-CV) as the most suitable for the gridding procedure among other RBFs." If HO-CV was used for parameter optimization, clarify this. Listing specific scores may not be necessary in the abstract.

Dear reviewer, we partly agree with your remarks and have rewritten this part slightly. However, including the scores' values in ESSD's paper abstracts seems quite common (e.g., https://doi.org/10.5194/essd-13-1273-2021, https://doi.org/10.5194/essd-16-3795-2024, https://doi.org/10.5194/essd-2024-586), so we decided not to remove them. We assume that this information could be valuable for the readers.

b) Introduction: Clearly state motivations, research questions, and approach. Avoid mixing conclusions into this section (e.g., lines 36–42). Eliminate redundancies (e.g., lines 43–45). Clarify if line 60 refers exclusively to observational datasets.

We have taken into account your suggestions and modified this section accordingly.

c) Reorganization: Consider moving Section 2.4 to the beginning of Section 3, avoiding redundancies (e.g., lines 156–157).

We have considered the suggestion and moved this section accordingly.

d) Section 4: Reassess why trend analysis is treated as an application while extremes analysis in Section 3 is treated as validation. Both seem to be dataset applications. Provide a clear strategy for distinguishing these analyses.

Thank you for this remark. We agree that extreme analysis in Section 3 should be treated as an application, as both are based on gridded datasets. We made this part the second subsection of Section 4, after the subsection on trend analysis.

5. Dataset Purpose: State whether this dataset is intended for near-real-time climate monitoring (e.g., with regular updates) or as a one-time release

   The dataset is mainly devoted to climate analysis and other climate-related applications. It is stressed in the introductory part and in the abstract.

6. Title. A more concise title, such as "PL1GD-T: High-Resolution Gridded Daily Air Temperature Dataset for Poland," is recommended.

   We have taken into account your suggestion and changed the title accordingly.

Minor comments:

Ensure consistent use of terms for TG, TN, and TX (e.g., TG is inconsistently referred to as TMEAN in Fig. 4 and TAVE in Section 4). Standardize abbreviations throughout the manuscript: e.g. replace 90% quantile and 10% quantile with 90th percentile and 10th percentile.

We have carefully checked the manuscript to avoid inconsistencies.

Line 260: This sentence seems abrupt and lacks context, making it feel disconnected from the surrounding text. Consider revising it to provide a smoother transition and explain its relevance to the discussion.

This line has been removed.

---

## Author Comment (AC2)

**RC2: 'Comment on essd-2024-433', Roberto Serrano-Notivoli, 03 Dec 2024**

Dear Roberto, thank you for your insightful comments and suggestions, which have significantly improved the clarity and rigour of our manuscript according to your review.

We have thoroughly revised our manuscript and addressed all the questions and concerns raised by the reviewer. Detailed responses are provided in blue following the reviewer's comments (in black).

The authors develop a gridded dataset of daily temperatures for Poland covering the 1951-2020 period, based on observed data from 347 stations. The research is oriented to the validation of the dataset. A very brief example of use is shown at the end of the manuscript.

While the overall objective of constructing a new daily dataset of temperatures for Poland is justified and well contextualized in the introduction, there are several unclear parts regarding the utilized data and the methodological approach.

Based on the reviewers' comments, we have extended the Summary section with an outlook part on planned improvements in subsequent versions of the dataset within the presented methodology.

For instance, regarding the **original data from stations**, a complete characterization of the raw temperature dataset is needed: Where are located the stations? (maybe a map would be useful);

The topographical map of Poland (Figure 1) is added with the stations' locations, and the data description has been extended in section 2.1.

Do all of them have daily data or they come from hourly information? Are they automatic or manual stations?

TN and TX are measured directly. A detailed description of the TG calculation procedure has been added to Eq.1. Manual and automatic station data are used in our study.

How many years of data (and gaps) they have? How is the temporal evolution of data availability?

We have conveyed it at the end of subsection 2.1 by commenting on Figure 2a, which shows daily data availability.

Did you apply any quality control/homogenization/gap filling procedure? This information and a basic statistical characterization are basic to contextualize the starting point of the grid and to understand the results.

Figure 2b distribution of available data percentage in the period has been added and commented on at the end of subsection 2.1. Additionally, Table 3 describes altitude-dependent station network characteristics.

In order to be clearer in the **methodological section**, the use of RBF must be further justified with more than a few references (Lines 105-108), mostly not related to temperature. While RBF is not particularly wrong, it is known as a "conservative" interpolation procedure, meaning that it usually reduces the spatial variability of the results which could be a problem in complex orography areas. As an example, the noted underestimation of higher values and overestimation of lower ones, can be attributed, with high confidence, to the interpolation scheme. Same situation can be observed at high elevations. Although higher RMSD values can be related to the scarcity of observations at those altitudes, the RBF is also probably

related. None of the cited and well-known datasets used this approximation. Although it can be perfectly valid for this dataset, it must be supported by a justification.

As you pointed out, we acknowledge the "conservative" nature of RBF and the potential for underestimating high values and overestimating low values. To address this, we have significantly expanded the methodological section (subsection 2.2) to provide a more thorough justification for our choice of RBF.

In addition, a gridded dataset must include a measure of uncertainty for all estimates to evaluate the reliability of the data at each location and time step.

Thank you for your valuable comment. Though including the measures in the datasets published through ESSD is not common, it should be considered as it allows the end user to assess the dataset's quality. As our dataset is not intended to be a one-time release and will get quasi-operational production, we will include the uncertainties in future releases. More detailed information on further steps in the dataset development is included in an additional paragraph of the Summary section.

Lastly, there is not a discussion section in which the new gridded dataset could be compared to others covering the region and cited in the introduction.

We have compared essential characteristics of PL1GD-T and E-OBS datasets in Table 6.

Apart from a general recommendation of a professional proofreading of English in the document, due to some confusing expressions, here are the minor comments, line by line:

**Introduction:**

L31: This statement is not valid for non-European regions.

We have removed 'European' from this sentence.

L36: Which is the rationale to choose this spatial resolution?

We have addressed this question in the last paragraph of the Introduction.

L71: Actually, there is an operational product starting in 1960 that is updated regularly: https://www.meteoswiss.admin.ch/dam/jcr:818a4d17-cb0c-4e8b- 92c61a1bdf5348b7/ProdDoc\_TabsD.pdf

Thank you for the clarification. This part of the section has been rewritten to include up-to-date information.

**Data and methods:**

L89: "IMGW-PIB". What is this?

The abbreviation states for Polish NHMS. It is added to the Introduction

L101-103: This means that all of the stations had hourly data?

Generally, SYNOP data consists of 8 hourly manual readings, KLIMAT consists of 4 hourly readings, and after automation, 10-minute measurements are available for both types of stations. We removed this sentence to make this part to avoid unnecessary information and clarified the procedure description in this subsection.

L120: How many "m" points you used? Did you set a radius of search? This is important since the availability of stations is not the same throughput the temporal period, and it has an impact on the estimation.

Thank you for raising this critical issue. m corresponds to the number of stations and varies temporally according to daily data availability. The RBFS do not introduce a radius for the search parameter. Instead, some of the functions contain a shape parameter. Section 2.2 was extended to describe the interpolation methodology in more detail.

L132-133: If I understood, you're assigning the value of a small 100x100m elevation pixel to the 100 pixels overlapped by a 1km2, right? The problem with this is that is in those areas with high elevation variability you are assigning a non-representative value to the larger pixel and that can lead to significant biases in temperature estimates. A correct approach would be using the mean or the median elevation of all 100x100 pixels overlapped by one 1x1km pixel.

**Thank you for raising this issue. We didn't express the approach fully. It is clarified in the extended description in that paragraph.**

L139-141: While this is valid to evaluate daily estimates, it cannot be used to evaluate long term trends or even monthly or annual aggregates because, for the comparison between a single pixel and their overlapping observations, you have non-continuous data or even data from different stations. To avoid this issue, it is usual to separate some complete data series (for example 20-30% of the total) and use them to validate the estimates at those locations. In addition, how did you randomly select the 5%? It should be a spatially driven randomization to avoid spatial biases in selecting stations. This procedure is implemented in almost all GIS software.

Thank you for the comment. We agree with your remark that the results of HO-CV cannot be used for climatological evaluation. As stated in that paragraph, HO-CV is only used to select the optimal RBF. As the 5% randomly selected stations vary daily, the interpolation is elaborated on the same 95% locations for all RBFs for a particular day.

**Results and discussion:**

L155-158: This is already stated in previous section

These sentences have been removed from that section.

L170 (Figure 1): What is the meaning of coloured lines? Please extend the figure caption.

The captions for the Figures presenting Taylor's diagrams are extended.

L193-195: Any interpretation for this? I guess that the RBF is smoothing the extremes.

We have commented on that in the Summary and outlook

L196 (Table 3): What is "cRMSD"? About Q95D and Q05D, I guess that they are the difference between observed and estimated Q95 and Q05, respectively. Please extend the table caption to make clear the meaning of all acronyms and the units in which the values are expressed.

cRMSD is the centred root mean squared difference defined by Eq. (13). The description is added to the table for clarification.

L225 (Table 4): At this point, the ratio of means or the ratio of standard deviations would be a better test, since the RMSD is an absolute value and we can't see here if there is a bias related to an over- or under-estimation by altitudes.

We agree with your statement. We have introduced RM and RSD in the altitude-dependent validation and removed parts related (text, table and figures) to RMSD.

Figures 3, 4 and 5: The tonal variation in a single-color scale avoids a correct interpretation. I recommend using a sequential color scale.

These figures are removed according to previous explanations.

L261: Just a conceptual note: you can derive past temperature variability, but this is not (or not necessarily) related to climate change.

This sentence has been removed from the manuscript.

L267: These values are surprisingly high. Please clarify if you are showing the average 90th percentile of TX for both periods or the absolute maximum one.

In this part, we compare the spatial distributions of TX. We showed the lowest and highest values in two periods. The lower value increased from -1.4 to 1.1. The values do not have to originate from the same grid point.

L290-291: This is not results and it should be removed.

Thank you for the comment. As suggested by the first reviewer, this part has been moved to section to section '4 Example application of the datasets' as subsection '4.2 Extreme analysis'

L315 (Table 5): Please, state if these values are the average of all the pixels.

The statement is already in the manuscript. It is added to the table's caption for clarification.

**Example of application:**

L320: Trends significance is not shown. In addition, when you say "selected stations", do you mean the pixels overlapping those cities?

We have added trend significance to the figure. Yes, the comparison is made between observed data and interpolated values from the nearest node. The sentence is corrected for clarification.

L322-323: This is not a fair comparison since you used the data from stations to build de gridded dataset.

Thank you for the comment, but we cannot agree with your statement. This is the application part, and height-dependent validation has been done in the appropriate section. Our motivation was to show that even in the case of high biases, the dataset is usable for trend analysis.

---

## Author Response (AR2)

**Report #1 Submitted on 05 May 2025**

**Anonymous referee #1**

Dear Authors, am satisfied with the changes you've made to the manuscript—thank you for addressing my comments. While re-reading the text, I noticed a few minor typos. For example, on line 85, "provide a high and spatial resolution" should be "provide a high spatial resolution," and on line 55, "As seen in Table 2 shows…" appears to be a grammatical error. Please take a moment to carefully proofread the manuscript to ensure the text is as clear and polished as possible. This is my final recommendation, and I look forward to seeing your paper published.

Dear referee,

Thank you for your comments and suggestions. We proofread the manuscript for any grammatical errors and typos. We corrected the mistakes you noticed and others we saw in the proofreading. We changed text formatting in several places and removed double spaces. A missing reference to Figure 1 was added, and its placement in the text was adjusted.

Dear Graciela,

We appreciate the chance to share our work with a broader audience. We are committed to enhancing our manuscript based on the valuable feedback and suggestions provided by the referees. Thank you and the support team for guidance throughout the publication process.

Best regards,
Adam Jaczewski, on behalf of the authors' team